# DETERMINISTIC TRAINING OF GENERATIVE AUTOENCODERS USING INVERTIBLE LAYERS

**Gianluigi Silvestri***
OnePlanet Research Center, imec-the Netherlands
Donders Institute for Brain, Cognition and Behaviour
`gianluigi.silvestri@imec.nl`

**Daan Roos***[†]
UvA-Bosch Delta Lab
`d.f.a.roos@uva.nl`

**Luca Ambrogioni**
Donders Institute for Brain, Cognition and Behaviour
Radboud University
`l.ambrogioni@donders.ru.nl`

## ABSTRACT

In this work, we provide a deterministic alternative to the stochastic variational training of generative autoencoders. We refer to these new generative autoencoders as AutoEncoders within Flows (AEF), since the encoder and decoder are defined as affine layers of an overall invertible architecture. This results in a deterministic encoding of the data, as opposed to the stochastic encoding of VAEs. The paper introduces two related families of AEFs. The first family relies on a partition of the ambient space and is trained by exact maximum-likelihood. The second family exploits a deterministic expansion of the ambient space and is trained by maximizing the log-probability in this extended space. This latter case leaves complete freedom in the choice of encoder, decoder and prior architectures, making it a drop-in replacement for the training of existing VAEs and VAE-style models. We show that these AEFs can have strikingly higher performance than architecturally identical VAEs in terms of log-likelihood and sample quality, especially for low dimensional latent spaces. Importantly, we show that AEF samples are substantially sharper than VAE samples.

## 1 INTRODUCTION

Variational autoencoders (VAEs) (Kingma & Welling, 2014; Rezende et al., 2014) have maintained an enduring popularity in the machine learning community in spite of the impressive performance of other generative models (Goodfellow et al., 2014; Karras et al., 2020; Van Oord et al., 2016; Van den Oord et al., 2016; Salimans et al., 2017; Dinh et al., 2014; Rezende & Mohamed, 2015; Dinh et al., 2017; Kingma & Dhariwal, 2018; Sohl-Dickstein et al., 2015; Nichol & Dhariwal, 2021). One key feature of VAEs is their ability to project complex data into a semantically meaningful set of latent variables. This feature is considered particularly useful in fields such as model-based reinforcement learning, where temporally linked VAE architectures form the backbone of most state-of-the-art world-models (Ha & Schmidhuber, 2018b;a; Hafner et al., 2020; Gregor et al., 2019; Zintgraf et al., 2020; Hafner et al., 2021). Another attractive feature of VAEs is that they leave ample architectural freedom when compared with other likelihood-based generative models, with their signature encoder-decoder architectures being popular in many areas of ML beside generative modeling (Ronneberger et al., 2015; Vaswani et al., 2017; Radford et al., 2021; Ramesh et al., 2021; 2022). However, VAE training is complicated by the lack of a closed-form expression for the log-likelihood, with the variational gap between the surrogate loss (i.e. the ELBO) and the true log-likelihood being responsible for unstable training and, at least in non-hierarchical models, sub-optimal encoding and sample quality (Hoffman & Johnson, 2016; Zhao et al., 2017; Alemi et al., 2018; Cremer et al., 2018; Mattei & Frellsen, 2018). Consequently, a large fraction of VAE research is devoted to tightening the gap between

---

*Equal contribution.
[†]Work done while at Radboud University

the ELBO and the true likelihood of the model. Gap reduction can be achieved both by devising alternative lower bounds (Burda et al., 2015; Bamler et al., 2017) or more flexible parameterized posterior distributions (Rezende & Mohamed, 2015; Kingma et al., 2016). Normalizing flows (NF) are deep generative models comprised of tractably invertible layers, whose log-likelihood can be computed in closed-form using the change of variable formula (Kobyzev et al., 2020; Papamakarios et al., 2021). However, this constraint appears to be at odds with autoencoder architectures, which map all relevant information in a latent space of different (often smaller) dimensionality. This is potentially problematic since naturalistic data such as images and speech waveforms are thought to live, at least approximately, in a lower dimensional manifold of the ambient space (Bengio et al., 2013; F. et al., 2016; Pope et al., 2021). It is therefore common to use hybrid VAE-flow models that deploy NFs for modeling the VAE prior and/or the variational posterior (Rezende & Mohamed, 2015; Kingma et al., 2016). However, training these models is often a delicate business as changes in the encoder and the prior cause a misalignment from the posterior, increasing the gap and causing a shifting-target dynamic that introduces instabilities and decreases performance. For this reason, the complex autoregressive or flow priors common in modern applications are often trained ex-post after VAE training (Van Den Oord et al., 2017; Razavi et al., 2019). In this paper we introduce a new approach for training VAE-style architectures with deterministically encoded latents. The key insight is that we can formulate an autoencoder within a conventional invertible architecture by using invertible affine layers and by keeping track of the deviations between data and predictions. Importantly, this can be done while leaving complete freedom in the design of the encoder, decoder and prior, which makes our approach a drop-in replacement for the training of existing VAE and VAE-style models. The resulting models can be trained by maximum-likelihood using the change of variables formula. We denote these new generative autoencoders as *autoencoders within flows* (AEF), since the autoencoder architecture is constructed inside a NF architecture.

## 2 PRELIMINARIES

In this section, we will outline the standard theory behind probabilistic generative modeling and non-linear dimensionality reduction. Consider a dataset comprised of data-points $x \in \mathbb{R}^N$. We refer to $\mathbb{R}^N$ as the ambient space. The dataset is assumed to be sampled from a $D$-dimensional curved manifold $\mathcal{M}$ embedded in the ambient space. We refer to $\mathcal{M}$ as the signal space. The dimensionality of the signal space reflects the true dimensionality of the signal while the dimensionality of the ambient space depends on the particularities of the measurement device (e.g. the nominal resolution of the camera).

**Variational autoencoders:** VAEs are deep generative models in which the density of each data-point depends on a $D$-dimensional stochastic latent variable $z \in \mathbb{R}^D$, which parameterizes the signal space. The emission model is often assumed to be a diagonal Gaussian with parameters determined by deep architectures: $p(x \mid z_j; \theta) = \mathcal{N}(x_j; f(z; \theta), f_s(z; \theta))$ , where $\theta$ denotes the model parameters. In this formula, the parameterized functions $f(z; \theta)$ and $f_s(z; \theta)$ are the outputs of a decoder architecture. The emission model is paired with a prior $p_0(z; \theta)$ over the latents. While the marginal likelihood is intractable, it is possible to derive a lower bound (the ELBO) by introducing a parameterized approximate posterior defined by the encoder architectures $g_m(x; \psi)$ and $g_s(x; \psi)$, which respectively return the posterior mean and scale over the latent variables. An additional normalizing flow transformation $n_{\text{post}}^{-1}(\cdot; \psi)$ is often included in order to account for the non-Gaussianity of the posterior (Kingma et al., 2016). Stochastic estimates of the gradient of the ELBO can be computed by expressing samples from the posterior as a differentiable deterministic function of the random samples (Kingma & Welling, 2014; Rezende et al., 2014). For our purposes, it is important to notice that the Gaussian reparameterization formula:

$$z(x, \epsilon; \psi) = g_m(x; \psi) + g_s(x; \psi) \odot \epsilon \,, \tag{1}$$

defines an affine invertible layer, formally analogous to those used in RealNVPs and related NFs (Dinh et al., 2017; Papamakarios et al., 2017). In the simplified case of Gaussian residual model with variance $\sigma^2$, the reparameterization of the ELBO leads to the following surrogate objective function

for one data-point $x$:

$$
\mathcal{L}_{\text{VAE}}(\theta, \sigma^2, \psi) = \mathbb{E}_\epsilon \left[ \underbrace{\frac{1}{2\sigma^2} \big\| x - f\big(z(x, \epsilon; \psi); \theta\big) \big\|_2^2}_{\text{Reconstruction error}} - \underbrace{\log p_0(z(x, \epsilon; \psi); \theta)}_{\text{Prior loss}} \right] \tag{2}
$$
$$
+ \frac{N}{2} \log 2\pi\sigma^2 - \underbrace{\mathcal{H}[q]}_{\text{Posterior entropy}} .
$$

While this loss has an interpretable appeal, it is important to keep in mind that it is just a tractable surrogate for the log-likelihood. The gap between the ELBO and log-likelihood can be decomposed into an inference gap and an amortization gap (Cremer et al., 2018). Large gaps lead to highly sub-optimal training since the network is trained on a loose approximation of the true objective.

**Normalizing flows and affine layers:** The unavailability of a closed form log-likelihood introduces a sub-optimality in VAE training that can only be remedied using complex posterior models or high variance lower bounds. NFs are alternative methods that assume the latent manifold to have the same dimensionality of the ambient space. Consider $\mathcal{M} = \mathbb{R}^N$ and $\phi(y; \theta)$ being an invertible differentiable mapping. Under these assumptions, the log-likelihood of the model can be expressed in closed form using the change-of-variable formula $\log p(x) = \log p_0(\phi^{-1}(y)) + \log \big| \det D\phi^{-1}(y) \big|$, where $D\phi^{-1}(y)$ is the Jacobi matrix of the inverse mapping. If the base distribution $p_0$ is standard normal, the loss has the form

$$
\mathcal{L}_{\text{NF}}(\theta) = \frac{1}{2} \left( \phi^{-1}(x; \theta) \right)^2 + \log \big| \det D\phi^{-1}(x; \theta) \big| . \tag{3}
$$

NF architectures are designed by composing $K$ invertible layers $\phi_k$. Affine layers divide the variables in two blocks $y_k^1$ and $y_k^2$, one of which remain unchanged and the other is scaled and translated based on the first $\phi_k(y_k^2) = y_{k+1}^2 = s(y_k^1) \odot y_k^2 + m(y_k^1)$, where $s$ is an arbitrary positive-valued function and $m$ is an unconstrained function. When applied in the inverse direction, this layer becomes $(y_{k+1}^2 - m(y_{k+1}^1))/s(y_k^1)$, which "predicts away" the mean and variance of one block given the other. Using several of these layers with randomized partitions gradually removes statistical dependencies until at the last layer the resulting $y$ variable approximately matches the uncorrelated Gaussian target $p_0$.

## 3 METHOD

### 3.1 AUTOENCODING WITH NORMALIZING FLOWS

In this section we show how to define a deterministic generative autoencoder within a NF architecture. Given its similarity with autoencoders, we named this flow architecture as *autoencoder within flows* (AEF). For now, we assume $D \ll N$, namely that the manifold dimensionality of the signal is much lower than the ambient dimensionality. We will remove this assumption in the next sections. Let us begin by partitioning the input signal $x \in \mathbb{R}^N$ in two subsets $x^{1:D}$ and $x^{D+1:N}$, as commonly done for coupling layers in NFs. In this paper, we refer to $x^{1:D}$ and $x^{D+1:N}$ as core and shell variables respectively. As we shall see, the core variables are in 1-to-1 relation with the latents while the shell variables are (approximately) "predicted away" by the latents. We define a mean (shell) encoder $g_m(\cdot; \theta) : \mathbb{R}^{N-D} \to \mathbb{R}^D$ and a scale (shell) encoder $g_s(\cdot; \theta) : \mathbb{R}^{N-D} \to \mathbb{R}^D$. These two architectures are analogous to the mean and scale encoders of a VAE except for the fact that they only take the shell variables as input. We also define an invertible *core encoder* $n^{-1}(\cdot; \theta) : \mathbb{R}^D \to \mathbb{R}^D$ that takes care of encoding the core variables. Using these architectures and the core/shell partition, we encode the data (both core and shell) into the latent variables via an invertible affine transformation:

$$
z = g_m(x^{D+1:N}; \theta) + g_s(x^{D+1:N}; \theta) \odot n^{-1}(x^{1:D}; \theta) . \tag{4}
$$

In order to recover the data from the latent, we define a decoder $f^{D+1:N}(\cdot; \theta) : \mathbb{R}^D \to \mathbb{R}^{N-D}$. This decoder only needs to reconstruct the shell variables since the core variables can be recovered by inverting the core encoder. As far as the scale encoding is not zero, the encoding formula defines a surjective transformation between the ambient space $\mathbb{R}^N$ and the latent space $\mathbb{R}^D$. Such a transformation is not invertible since information concerning the shell variables can be lost and,

consequently, $f^{D+1:N}(z(x);\theta) \neq x$. We can circumvent this problem by retaining the residuals of the autoencoder as additional variables:

$$\delta^{D+1:N} = x^{D+1:N} - f^{D+1:N}(z;\theta) , \tag{5}$$

These variables retain the information lost during encoding, thereby completing the surjective encoding formula into an invertible transformation $\Phi^{-1} : \mathbb{R}^N \to \mathbb{R}^N$ defined as $\Phi^{-1}(x^{1:D}, x^{D+1:N}) \to z, \delta^{D+1:N}$. Formally, the variables $\delta^{D+1:N}$ are parts of the 'latents' of the NF. However, conceptually they are not true latent variables as they can only account for additive white noise. We can now define a base distribution $p_0(z;\theta)$ to the latents. This distribution defines a generative model in the latent space $\mathbb{R}^D$ and it is analogous to the prior in VAEs. For this reason, we will often (improperly) refer to $p_0(z;\theta)$ as the 'prior'. However, it is important to keep in mind that this distribution is not a Bayesian prior since AEFs latents do not have a straightforward Bayesian interpretation. Moreover, we define an error distribution $r(\delta;\theta)$ to the residuals. It is important for this distribution to have a learnable scale parameter that can be scaled down during training as the residuals are "predicted away". In summary, the invertible architecture $\Phi^{-1}$ is trained to map the distribution of the data into a factorized distribution of latent codes and residuals:

$$\Phi^{-1}(x^{1:D}, x^{D+1:N};\theta) \sim p_0(z;\theta)r(\delta;\theta) \tag{6}$$

The exact negative log-likelihood loss can be obtained by using the change of variable formula of normalizing flows (Eq. 3):

$$\mathcal{L}_{\text{AEF}}(\theta,\sigma^2) = \underbrace{\frac{1}{2\sigma^2}\left\|x^{D+1:N} - f^{D+1:N}\big(z(x;\theta);\theta\big)\right\|_2^2 + \frac{N-D}{2}\log 2\pi\sigma^2}_{-\log r(\delta;\theta)} - \log p_0(z(x;\theta);\theta)$$

$$\tag{7}$$

$$-\log|\det D\Phi^{-1}(x;\theta)| ,$$

where for the sake of simplicity we assumed (centered) Gaussian residual noise with (trainable) variance $\sigma^2$. It is now possible to note some striking similarities between the flow just described and a VAE. Eq. 4 resembles the Gaussian reparametrization trick for variational autoencoders, with the reparameterized posterior noise replaced by the core variables. Furthermore, equation 7 closely resembles the ELBO in equation 2 but without noise injection, with the reconstruction error applied only to the shell variables and the Jacobian term replacing the entropy of the posterior distribution. As usual in flows, the AEF architecture can be used as a generative model simply by sampling the latent code $z$ from the 'prior' $p_0(z;\theta)$ and inverting the AEF transformation $\Phi^{-1}(\cdot)$. Since it is usually not useful to add residual noise to the generated data, we can set $\delta^{D+1:N} = 0$ instead of sampling it from the error distribution. This results in the following procedure:

$$z \sim p_0(z;\theta) \;\to\; x^{D+1:N} = f^{D+1:N}(z;\theta) \;\to\; x^{1:D} = n\left(\frac{z - g_m(x^{D+1:N};\theta)}{g_s(x^{D+1:N};\theta)}\right) . \tag{8}$$

This is visualized in Figure 3 (b) in Appendix A, and implemented in Algorithm 2.

### 3.2 PARTITION STRATEGIES AND AMBIENT SPACE EXPANSION

So far, we did not specify how to select core and shell variables. The most straightforward approach is to to use an arbitrarily chosen partition of the original variables. For example, we could use a random partition or, in an image, we could extract a central sub-image of "core pixels" and have the image with cropped center as shell. An AEF with this kind of partition interpolates between a regular normalizing flow (for $D \approx N$) and an autoencoder (for $D \ll N$). However, this approach limits the number of latents to be smaller than $N$ and reduces compatibility with the VAE literature. Furthermore, in the case of noise corrupted data this partitioning strategy does not allow for denoising of the core variables since they do not have corresponding deviations and error distributions. All these issues can be circumvented if we appropriately expand the ambient space. In fact, the dimensionality of the ambient space is usually largely arbitrary, depending on factors such as the sampling resolution of cameras and digital microphones instead of the physical features of their respective measured signals. Consider a parameterized injective function $\Psi : \mathbb{R}^N \to \mathbb{R}^{N+D}$, defined as follows $\Psi(x;\gamma) = (x, w = h(x;\gamma))$, where $\gamma$ are the transformation parameters. This function expands the ambient dimensionality but, since the transformation is injective and deterministic, it leaves the dimensionality of the signal

manifold unchanged. Conceptually, this should be seen as a form of feature expansion and not an architectural component of a flow. In this expanded space, we can define deviations for all the original variables and, consequently, use a loss with VAE-style reconstruction error on all variables. This can be done simply by using a regular AEF architecture with the original variables $x$ as shell and $w = h(x)$ as core. This results in the following encoding layer:

$$z = g_m(x;\theta) + g_s(x;\theta) \odot n^{-1}(h(x;\gamma);\theta) . \tag{9}$$

Note that this differs from the well-known VAE reparameterized encoding formula (E. 1) just by the fact that the input white noise is replaced by a deterministic function of the data. This is the only architectural difference between VAEs and AEF in the extended space. It is not immediately obvious that the feature expansion parameters $\gamma$ can be trained by maximizing the log-likelihood. However, this joint training can be fully justified as the minimization of (the limit of) KL divergences (see Section 3.3), which results in the following objective function:

$$\mathcal{L}_{\text{AEF}}(\theta, \sigma^2, \gamma) = \frac{1}{2\sigma^2}\big\|x - f\big(z(x, h(x;\gamma);\theta);\theta\big)\big\|_2^2 + \frac{N}{2}\log 2\pi\sigma^2 \tag{10}$$
$$- \log p_0(z(x, h(x;\gamma);\theta);\theta) - \log|\det D\Phi^{-1}(x, h(x;\gamma);\theta)| ,$$

which is just the log-likelihood in the expanded ambient space. We include the pseudocode for computing the objective function as negative log likelihood in Algorithm 1.

---

**Algorithm 1** Negative Log Likelihood AEF on expanded ambient space: $g$: Encoder; $f$: Decoder; $h$: feature expansion map; $n$: core encoder; $p_0$: Base distribution ('prior'); $r$: error distribution; $\theta$: model parameters, $\gamma$: feature expansion parameters; $x$: input image

> **procedure** NEGATIVELOGLIKELIHOOD($x$)
>     $\log|\det J| \leftarrow 0$
>     $w \leftarrow h(x;\gamma)$
>     $z \leftarrow g_m(x;\theta) + g_s(x;\theta) \odot n^{-1}(w;\theta)$
>     $\log|\det J| \leftarrow \log|\det J| + \log|\det J(n^{-1}(w;\theta))| + \sum \log g_s(x;\theta)$
>     $\delta \leftarrow x - f(z;\theta)$
>     **return** $-(\log p_0(z;\theta) + \log r(\delta;\theta) + \log|\det J|)$

---

During generative sampling, the additional variables $w$ are redundant and can be discarded. This results in a generative sampling formula that is identical to the one used in VAEs (see also Algorithm 3): $x = f(z;\theta),\ z \sim p_0(z;\theta)$. Consequently, the core flow $n$ does no longer participate in the generative sampling and it instead has an auxiliary role. This is a sign of its strong relation with the posterior flow in a VAE, which is already implicit in Eq. 9. The marginal likelihood in the original ambient space $p(x) = \int p(x, w)\mathrm{d}w$ cannot be obtained in closed-form from the joint $p(x, w)$. Note that, at least from the point of view of manifold learning and dimensionality reduction, the original likelihood $p(x)$ itself does not have a privileged interpretation as the ambient dimensionality is usually arbitrary and the 'true' likelihood of the data is degenerate and lives on a lower dimensional manifold. Nevertheless, $p(x)$ is often useful for evaluation purposes (model comparison). Therefore, we provide an efficient importance sampling scheme in Appendix C. All likelihood results reported in this paper have been corrected using this method so as to be comparable with the VAE baselines.

---

**Algorithm 2** Sampling AEF with core-shell partition: $g$: Encoder; $f$: Decoder; $n$: core encoder; $p_0$: base distribution ('prior'); $\theta$: model parameters

> **procedure** SAMPLE
>     $z \sim p_0(z;\theta)$
>     $x^{D+1:N} \leftarrow f(z;\theta)$
>     $x^{1:D} \leftarrow n\left(\frac{z - g_m(x^{D+1:N};\theta)}{g_s(x^{D+1:N};\theta)}\right)$
>     $x \leftarrow cat(x^{1:D}, x^{D+1:N})$
>     **return** $x$

---

**Algorithm 3** Sampling AEF with expanded space: $f$: Decoder; $p_0$: base distribution ('prior'); $\theta$: model parameters

> **procedure** SAMPLE
>     $z \sim p_0(z;\theta)$
>     $x \leftarrow f(z;\theta)$
>     **return** $x$

---

### 3.3 TRAINING THE EXPANSION PARAMETERS $\gamma$

The feature expansion map $h(x, \gamma)$ is, strictly speaking, not a component of the flow architecture. Therefore, in spite of its intuitive appeal, it is not obvious that we can train $\gamma$ by minimizing the joint negative log-likelihood. However, this can be fully justified as minimization of the KL divergence between the generative model and the empirical distribution of the data with expanded dimensionality. The idea is to define a loss functional that can be used to make two parameterized distributions approach each other, instead of just having a parameterized distribution approaching a fixed target distribution. We will start considering a stochastic dimensionality expansion and then show that the deterministic case can be obtained as a limit. We will denote the empirical sampling distribution of the dataset as $d(x)$. Now consider the joint distribution of the empirical data and the stochastically expanded latent dimensions:

$$q_\sigma(x, w) = d(x)\mathcal{N}(h(x, \gamma), I\sigma) \tag{11}$$

where the expansion is performed using a Gaussian conditional density with fixed variance. The KL divergence between the expanded empirical distribution $q(x, w)$ and the joint distribution of the AEF model $p(x, w)$ is given by:

$$D_{\mathrm{KL}}\left(q_\sigma(x, w), p(x, w)\right) = -\mathbb{E}_{x, \epsilon \sim q}[\log p(x, w)] + c \,, \tag{12}$$

where the constant $c$ is the differential entropy of $q_\sigma(x, w)$, which is independent from both $\theta$ and $\gamma$. This happens because the differential entropy of a Gaussian variable does not depend on its mean. We can now compute the gradient with respect to $\gamma$:

$$\nabla_\gamma D_{\mathrm{KL}}\left(q_\sigma(x, w), p(x, w)\right) = -\mathbb{E}_{x, \epsilon}[\nabla_\gamma \log p(x, w)] \,, \tag{13}$$

which, up to a constant, is just the gradient of the stochastically augmented joint log-likelihood. We can now notice that, while the KL divergence is not well-defined at the limit $\sigma \to 0$, the limiting gradient is well-defined since the diverging entropy term does not depend on $\gamma$. This leads us with the following limiting gradient:

$$\lim_{\sigma \to 0} \nabla_\gamma D_{\mathrm{KL}}\left(q_\sigma(x, w), p(x, w)\right) \tag{14}$$

$$= -\lim_{\sigma \to 0} \mathbb{E}_{x, \epsilon}[\nabla_\gamma \log p(x, h(x, \gamma) + \sigma \odot \epsilon)] \tag{15}$$

$$= -\mathbb{E}_x[\nabla_\gamma \log p(x, h(x, \gamma))] \,, \tag{16}$$

where $h(x, \gamma) + \sigma \odot \epsilon$, in the second line, is the the reparameterization formula for the conditional Gaussian variable $w$. We can now see that, up to a proportionality constant, is the gradient of the negative log-likelihood used to train the flow.

## 4 RELATED WORK

**Improvements of VAE training**: In recent years, there have been many works analyzing and improving the training procedure of VAEs. Works such as (Hoffman & Johnson, 2016), (Zhao et al., 2017) and (Alemi et al., 2018) diagnose some issues with ELBO training and propose a series of methods to improve results. (Rezende & Viola, 2018) and (Dai & Wipf, 2019), propose an augmented objective to tackle specific optimization issues. In the same vein, (Cremer et al., 2018) and (Mattei & Frellsen, 2018) offer a theoretical analysis of the factors affecting the quality of approximate inference in VAEs and introduce strategies to alleviate them. None of these works introduce an exact maximum likelihood training. A popular way to improve VAEs is to use a more flexible prior or posterior distribution. (Rezende & Mohamed, 2015) and (Kingma et al., 2016) use NFs as variational posteriors, showing how the aggregate posterior matches the prior more closely, while (Tomczak & Welling, 2018) use a variational mixture of posteriors as prior. (Morrow & Chiu, 2020) use a two-stage training procedure, in which first they train a VAE with a NF prior, and then combine the decoder with Glow (Kingma & Dhariwal, 2018) for improved sample quality. An alternative line of research is the use of hierarchical priors with several stochastic layers (Sønderby et al., 2016; Maaløe et al., 2019). More recent works use hierarchical VAEs with many layers, achieving state of the art log-likelihood and generating images with impressive sample quality (Vahdat & Kautz, 2020; Child, 2021; Hazami et al., 2022). These works use a latent dimensionality that is greater, often by orders of magnitude, than the dimensionality of the ambient space. This dimensionality expansion greatly ameliorate the problem of posterior separations diagnosed in (Dai & Wipf, 2019) at

the price of higher computational and memory costs and lower interpretability. Interestingly, a recent post-training analysis showed that just a few percents of the hierarchical VAE's latent dimensions are needed to encode the data (Hazami et al., 2022).

**Stochastic auxiliary variables and augmented normalizing flows:** The fact that VAE encoding can be seen as a form of affine coupling was first noticed in (Dinh et al., 2014) (Appendix C). This allows one to conceptualize the one-sample reparameterization estimate of the ELBO as a likelihood in a stochastically augmented space. However, this is just an equivalent re-formulation and does not solve the inference sub-optimality problems of VAEs, which in this setting can be explained by the fact that sharp posteriors correspond to singular points of non-invertibility. This approach was recently generalized in (Huang et al., 2020), where several VAE-style affine layers are stacked in an flow architecture that takes noise augmented input. The stacking procedure removes the signal/noise separation and autoencoder-like loss of VAEs since the final prediction is not compared with the original image but only with the previous layer. The resulting model is very similar to other auxiliary-augmented flows (Cornish et al., 2020; Weilbach et al., 2020; Caterini et al., 2021a).

**Two-steps training procedures**: A recent trend in generative modeling is to disentangle the tasks of learning a low-dimensional latent representation and maximum-likelihood density estimation. (Dai & Wipf, 2019) proposes a two-stage procedure for training VAEs, and show its effectiveness both theoretically and empirically. (Ghosh et al., 2020) propose to instead use an explicit regularization scheme for the decoder, and employ an ex-post density estimation on the latent space to allow for sampling and ensure that the latent space is distributed according to a simple distribution. Similarly, (Xiao et al., 2019) and (Böhm & Seljak, 2020) use a deterministic autoencoder to learn the latent representation of the data, and a normalizing flow to model the distribution of such latents, leading to better density estimation while avoiding over-regularization of the latent variables. More recently, (Loaiza-Ganem et al., 2022) discusses the problem of manifold overfitting, which arises when the manifold is learned but not the distribution on it, and propose a two-step training procedure applicable to all the likelihood-based models.

**NFs on manifolds**: Several authors propose variations of NFs that can model data on a manifold. Works such as (Gemici et al., 2016), (Mathieu & Nickel, 2020) and (Rezende et al., 2020) assume that the dimensionality reducing map is already known and available. (Kim et al., 2020; Horvat & Pfister, 2021) propose methods based on adding noise to the data, which are capable of learning unknown lower-dimensional manifolds. Injective flows strive for the same goal by using injective deterministic transformations to map the data to a lower dimensional base density. (Brehmer & Cranmer, 2020), (Kothari et al., 2021) and (Cramer et al., 2022) train injective flows with a two-steps training procedure in which they alternate manifold and density training, while (Kumar et al., 2020) introduces lower bounds on the injective change of variable. (Cunningham et al., 2020) combines injective flows with an additive noise model to account for deviations from the learned manifold using stochastic inversions trained on a variational bound. In (Caterini et al., 2021b) the authors train injective flows by evaluating the in-manifold likelihood using a modification of the change of variable formulas for rectangular Jacobi matrices, which is then supplemented by an additional ad-hoc reconstruction loss. (Ross & Cresswell, 2021) uses conformal embeddings to train injective flows with exact on-manifold log-likelihood plus an additional *ad hoc* reconstruction loss.

**SurVAE flows**: A work that is in spirit similar to ours is SurVAE (Nielsen et al., 2020), in the sense that it aims at bridging the gap between Normalizing Flows and VAEs by combining their strengths. SurVAEs extend flow models by incorporating stochastic and surjective transformations in both the generative and the inference direction. The (generative direction) surjective transformations give rise to stochastic estimates of the likelihood contribution and introduce lower bound likelihood estimates. On the other hand, surjections in the inference direction allow exact probability computations for the latent variables. However, these inference surjections themselves cannot be straightforwardly used to perform generative dimensionality reduction since the stochastic forward transformation $p(x|z)$ is intractable and depends on the data probability itself via Bayes theorem (which is exactly the quantity we want to estimate in generative modeling). In general, the contributions of our work and SurVAE are almost orthogonal and could be combined to obtain the benefits of both techniques, for example by incorporating surjections in our decoder architecture so as to enforce symmetries in the data.

## 5 EXPERIMENTS

We now show empirically that, at least in complex naturalistic datasets such as CelebA-HQ and ImageNet Karras et al. (2018); Deng et al. (2009), the exact maximum likelihood training leads to drastically improved results compared to architecturally equivalent VAEs. Our aim is to show the difference in performance between exact log-likelihood and ELBO objective functions when the architecture is kept constant. Therefore, in our VAE baselines we do not use the many regularization, annealing and KL scaling terms that are somewhat common in VAE applications (Vahdat & Kautz, 2020; Child, 2021). However, we make sure to compare architecturally identical models with strong prior and variational posteriors, which in themselves ameliorate many of the pathologies of the ELBO (Kingma et al., 2016; Tomczak & Welling, 2018). As main baseline, we use the original VAE algorithm as introduced in (Kingma & Welling, 2014) with an Inverse Autoregressive Flow (IAF) posterior and a Masked Autoregressive Flow (MAF) prior. Apart from the feature expansion layer, this baseline is architecturally identical to our AEF (linear) model, which uses learnable linear features as core variables. We also test two AEF models that do not expand the ambient space, one with the center pixels and the other with the corner pixels as core variables, referred to as AEF (center) and AEF (corner) respectively. Additional experiments on denoising are reported in Appendix B, while more details about the experiments and results can be found in Appendix D and E. The code for all the experiments is available at: https://github.com/gisilvs/AEF.

**Generative modeling and manifold learning:** To compare the generative performance of AEFs with VAEs we test on CelebA-HQ resized to $64 \times 64$ and $32 \times 32$, and ImageNet resized to $32 \times 32$. We focus on models with significantly less latent variables than observable dimensions since this is the regime that is the most problematic for traditional VAE training. We use a complex Encoder-Decoder architecture with residual blocks, similar to the one used in (Child, 2021), and compare the performances of the AEF (with linear core variables) and its architecturally identical VAE for different latent dimensions. Table 1 shows that AEFs significantly outperform their architecturally equivalent VAEs both in terms of bits per dimension and FID score, generating significantly sharper and more detailed samples (Fig. 1, 2). To compute bits per dimension, we use importance sampling, as described in Appendix D.6. Additionally, we look at smaller scale datasets: MNIST, FashionMNIST and KMNIST (Deng, 2012; Xiao et al., 2017; Clanuwat et al., 2018). Here we use small conventional encoder and decoder architectures (see Appendix D.1). Results on MNIST are shown in Table 1 for a latent dimensionality of 2 and 32. In Appendix E we present these results for all latent dimensions and datasets, as well as the results for AEF (center) and AEF (corner). Performance for all three versions of AEF is comparable, with the linear AEF performing slightly better. Overall, we observe an increase in performance for AEFs compared with VAEs. On the other hand, VAE outperforms AEF for a high number of latent dimensions. In Appendix F we investigate the importance of the posterior and prior flows on generative performance for both AEFs and VAEs. Here we observe that incorporating a prior flow is very important to sample quality for both models. Additionally, we observed that, in this setting, even the AEF without flows performs better than a VAE with both posterior and prior flow in terms of BPD and FID.

| | | MNIST | | CelebA ($64 \times 64$) | | | CelebA ($32 \times 32$) | | | ImageNet | |
|---|---|---|---|---|---|---|---|---|---|---|---|
| Models | | 2 | 32 | 64 | 128 | 256 | 64 | 128 | 256 | 128 | 256 |
| VAE | BPD | 2.40 | **1.80** | 6.98 | 6.86 | 6.79 | 6.90 | 6.64 | 6.63 | 6.89 | 6.74 |
| | FID | 57.5 | **16.2** | 106 | 92.6 | 90.9 | 87.6 | 67.0 | 63.5 | 179 | 160 |
| AEF | BPD | **2.37** | 1.82 | **6.35** | **6.07** | **5.78** | **6.20** | **5.88** | **5.57** | **6.30** | **6.21** |
| | FID | **55.8** | 17.7 | **76.5** | **60.8** | **55.8** | **51.9** | **38.4** | **30.9** | **131** | **123** |

Table 1: Bits per dimension (BPD) and Frechet Inception Distance (FID) for AEF and VAE models trained on various datasets. The second row denotes the latent dimensionality used. Lower is better for both metrics. For MNIST we report the mean over five runs, for the other datasets we report the mean over two runs. For both VAE and AEF, BPD values are estimated using importance sampling. We refer to Appendix E for more results, and Appendix D.6 for details on the importance sampling estimators.

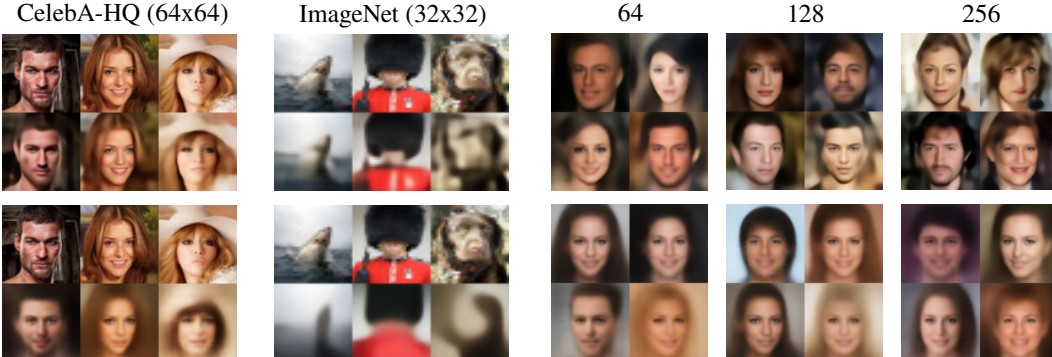

| CelebA-HQ (64x64) | ImageNet (32x32) | 64 | 128 | 256 |

Figure 1: Reconstructions from an AEF (top row) and VAE (bottom row) with equivalent architectures trained on rescaled CelebA-HQ and rescaled ImageNet with 256 latent dimensions.

Figure 2: Samples from an AEF (top row) and VAE (bottom row) with equivalent architectures trained on rescaled CelebA-HQ ($64 \times 64$) with 64, 128, and 256 latent dimensions respectively. Images were sampled with a temperature of $0.85$.

## 6 DISCUSSION AND CONCLUSIONS

In this work we showed that autoencoders, if properly constructed out of invertible layers, can be trained by maximum-likelihood either in the original ambient space or in an appropriately expanded space. This latter approach results in an objective function that can be directly used in the training of any pre-existing VAE and VAE-like model that uses Gaussian residual noise. In our experiments, we showed that in many datasets AEFs perform strikingly better than VAEs. Importantly, the AEF images where not affected by the blurriness typical of low-dimensional VAEs, which resulted in very remarkable difference in the quality of samples and reconstructions. This is an interesting and perhaps surprising result as the AEF and VAE models were architecturally identical. Given the arguments presented in Alemi et al. (2018), we conjecture that this difference is due to the failure of the VAEs to converge to a sufficiently sharp posterior, which has been proven to result in poor separation between the encoding of the training samples. We further conjecture that this failure is due to the fact that the optimum is very close to a singular point of non-invertibility of the VAE architecture, which results in numerical instabilities due to the diverging KL term. This problem does not affect AEFs since the encoding is always deterministic. In our experiments, the main exception to this trend was MNIST with 32 latent dimensions, where the VAE performed consistently better. We conjecture that this is due to the relatively large ratio between the latent dimensionality and the true signal dimensionality, which results in less concentrated (and less unstable) variational posterior distributions. In this regime, the extra statistical variability provided by the posterior samples of the VAE can loosen the topological constraints of the architecture, potentially leading to higher performance Cornish et al. (2020); Caterini et al. (2021a). When compared to VAEs, the main limitation of AEFs is that they cannot straightforwardly use discrete emission models (decoders) since their flow architecture is assumed to work on continuous data. Another limitation is that, since we are not using stochastic auxiliary variables, AEFs have the same topological constraints of regular NFs (Cornish et al., 2020; Caterini et al., 2021a). However, auxiliary variables can be straightforwardly added to AEFs using standard methods Caterini et al. (2021a). Finally, from the point of view of the NF literature, our work opens the door to hybrid autoencoder-flow models that can learn how to reduce the latent dimensionality by "predicting away" some dimensions, possibly at several different stages of processing. Adding manifold learning capabilities to NFs has great potential since it can avoid some of the pathologies of invertible models when used on lower dimensional data, while at the same time increasing training efficiency.

## 7 REPRODUCIBILITY

We provide access to all the code used in our experiments here: https://github.com/gisilvs/AEF. The readme contains instructions on how to reproduce the experiments presented in the paper from the command line, as well as a Jupyter notebook example on training an AEF from scratch using the

provided code. To further increase the clarity of our proposed method, we have added diagrams that visually explain the sampling and inference procedure for our model in Appendix A, as well as corresponding pseudocode (algorithms 1, 2, 3). The reader can find additional explanations of the methods in Sections 3.3 and C. Additional details on the performed experiments such as description of architectural details and hyperparameter used in this work can be found in Section D. Ablation experiments are found in Section F in the Supplementary material.

## ACKNOWLEDGEMENTS

OnePlanet Research Center aknowledges the support of the Province of Gelderland.

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

## SUPPLEMENTARY MATERIALS

## A   ADDITIONAL DIAGRAMS

In Figure 3, we show diagrams for inferece and sampling procedures, for AEFs with partitioning and expanded ambient space.

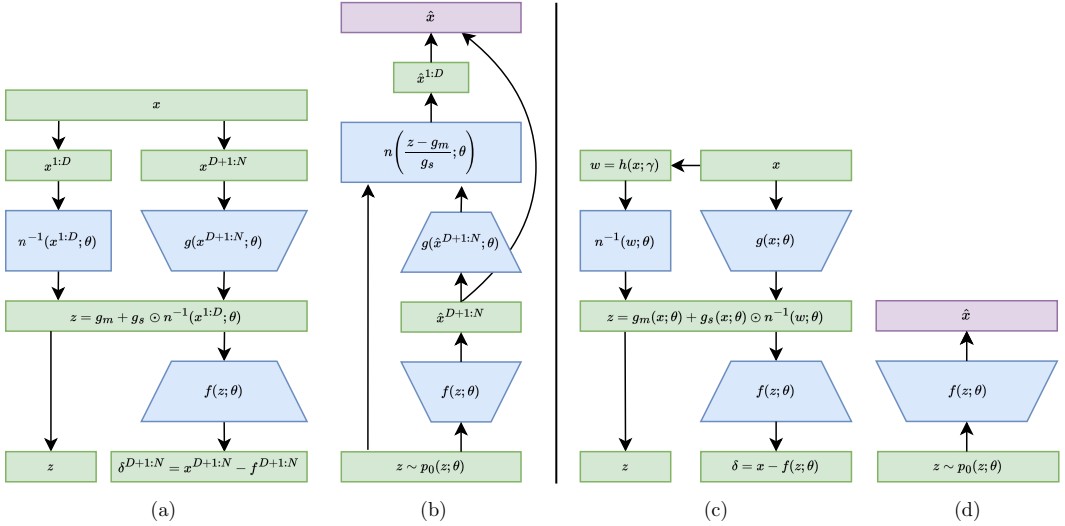

Figure 3: Diagrams showcasing the inference and sampling process for (a, b) AEFs with partioning; (c, d) AEFs with expanded ambient space.

## B   DENOISING EXPERIMENTS

Denoising is one of the main applications of the classical autoencoder literature. The basic idea is that the signal can be compressed in a low dimensional manifold while the (white) noise, being incomprehensible, is filtered out. The denoising problem shows the capacity of our training scheme to separate the signal and the noise spaces, thereby performing an adaptive non-linear filtering. We test performance on CelebA-HQ ($32 \times 32$), and the MNIST, FashionMNIST and KMNIST datasets with different noise levels. We compare against architecturally equivalent VAEs and least squares denoising autoencoders (AE). All models were trained exclusively on noisy data. To compare performance we look at the mean squared error between the Inception feature activations (the same ones as used for the FID score) for an original, noiseless sample and the reconstruction based on its noisy version. Results for CelebA-HQ and MNIST are shown in Table 2 and Fig. 4 for various noise regimes, while results for FashionMNIST and KMNIST and the other noise regimes can be found in Appendix E.2. For CelebA-HQ we see that the gap in performance between AEF and VAE for denoising is similar to the one in generative modeling. On the other hand, for the less complex MNIST datasets we observe that the VAE often performs better than both the AEF and the denoising AE, which suggests that the posterior uncertainty of the stochastic latent variables of the VAE may play a beneficial role in the denoising performance.

## C   IMPORTANCE SAMPLING FOR LINEAR AEF

After training, if we wish to compute the probability $p(x)$ for model comparison purposes, we would need to solve the integral $p(x) = \int p(x, w)dw$, which cannot be obtained in closed form. However, we can use an importance sampling scheme:

$$p(x) = \int p(x, w)dw = \int q(w)\frac{p(x, w)}{q(w)}dw \approx \frac{1}{K}\sum_{k}\frac{p(x, w_k)}{q(w_k)} \tag{17}$$

| | CelebA-HQ (128) | | | MNIST (2) | | | MNIST (32) | | |
|---|---|---|---|---|---|---|---|---|---|
| | 0.05 | 0.1 | 0.2 | 0.5 | 0.75 | 1.0 | 0.5 | 0.75 | 1 |
| AE | 0.080 | 0.078 | 0.078 | 0.135 | 0.142 | 0.144 | 0.086 | 0.115 | 0.133 |
| VAE | 0.082 | 0.081 | 0.085 | 0.085 | 0.094 | 0.109 | **0.045** | **0.058** | **0.075** |
| AEF | **0.058** | **0.063** | **0.075** | **0.082** | **0.091** | **0.102** | 0.053 | 0.072 | 0.094 |

Table 2: Mean squared error between inception feature activations of original inputs and reconstructions of noisy inputs. The number between parentheses denotes the latent dimensionality of the models. The second row gives the standard deviation of the noise distribution. For CelebA-HQ and MNIST we report the mean over two and five runs respectively.

|  (a) AEF | (b) VAE | (c) AE |
|---|---|---|

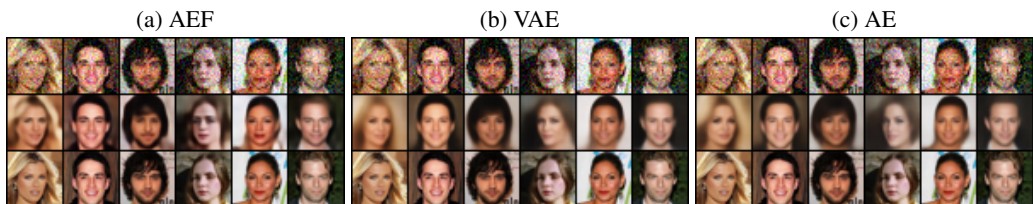

Figure 4: Examples of reconstruction performance of AEF, VAE, and deterministic autoencoder (AE) trained on CelebA samples with a noise level ($\sigma$) of 0.1, and a latent dimensionality of 128. **Top**: image with added noise; **Middle**: reconstructed image; **Bottom**: original image.

To do so, we first compute $w = h(x; \gamma)$, then take $K$ samples from a normal distribution centered around $w$: $w_{1,...,K} \sim q(w) = \mathcal{N}(w, \epsilon)$, where $\epsilon$ is the approximate posterior scale and needs to be tuned on the validation set for each model. We perform the rest of the computations using $w_{1,...,K}$ instead of $w$, and finally compute the probability in Eq. 17. To reduce the variance of the estimator, we additionally use importance weighted sampling:

$$\log p(x) \approx \log \mathbb{E}_{w_{1,...,K} \sim q(w)} \left[ \frac{1}{K} \sum_k \frac{p(x, w_k)}{q(w_k)} \right] \tag{18}$$

## D   EXPERIMENTS' DETAILS

### D.1   ENCODER AND DECODER

For the MNIST datasets, all the encoders and decoders consist of a two-layers convolutional neural network. The encoder uses $3 \times 3$ kernels, 64 for the first layer and 128 for the second. Each layer is followed by ReLU activation. Finally, two linear layers map the feature maps to mean and standard deviation of the latent space respectively. Similarly, the decoder first uses a linear layer to increase the dimensionality of the latent samples, and then two transposed convolutional layers with respectively 128 and 64 kernels of size 4. The decoder outputs two values: the mean and the standard deviation, as a set of trainable parameters constrained with softplus activation.

For the larger CelebA and ImageNet datasets we use an encoder and decoder with many more layers and residual blocks, an architecture similar to (Child, 2021). In particular, we use the same residual bottleneck block, but the encoder does not output activations at intermediate layers, and the decoder processes only the input coming from the previous layer, without stochastic sampling and prior computation. In other words, we reuse the residual bottleneck blocks from (Child, 2021) to build a "traditional" encoder-decoder architecture, and the outputs of the encoder and decoder are the same as for the manifold learning and denoising experiments. In all the experiments, we use four residual bottleneck blocks for each feature map size.

## D.2  MASKED AUTOREGRESSIVE FLOW

For all of our NFs, we use and adapt implementations from (Durkan et al., 2020). Our MAF models stack $K$ MADE autoregressive layers (Germain et al., 2015), each with 2 residual blocks with $N$ hidden units. We add ActNorm (Kingma & Dhariwal, 2018) between each autoregressive layer. IAF is simply the inverse of MAF. When MAF or IAF are used within an autoencoder architecture, whether as prior, posterior or encoder flow, we use $K = 4$ and $N = 256$.

## D.3  PREPROCESSING LAYER

For all models, we apply a preprocessing bijective transformation like the one used in (Papamakarios et al., 2017):

$$x = \text{logit} \left( \lambda + (1 - 2\lambda)z \right) \tag{19}$$

where $z$ is the input image and $\lambda$ is a parameter. We use $\lambda = 1e^{-6}$ for the MNIST-like datasets and $\lambda = 0.05$ for CelebA-HQ and ImageNet. This transformation is then followed by an ActNorm layer.

## D.4  DATASETS AND DATA PRE-PROCESSING

In all our experiments, we use a dequantized version of the data, in which we first add uniform noise $u \sim \mathcal{U}(0, 1)$ to the image and then divide by 256. Division by 256 requires an adjustment in the log likelihood, which we take into account when computing the bits per dimension.

For the denoising experiments we add gaussian white noise $\mathcal{N}(0, \sigma)$ to the samples, with different levels of $\sigma$ depending on the dataset: $0.25$, $0.5$, $0.75$ and $1.0$ for MNIST datasets, and $0.05$, $0.1$ and $0.2$ for CelebA-HQ. By varying the standard deviation of the noise distribution we can increase the intensity of the noise. After adding the noise to the images we clip them so that the pixel values stay in their original range.

For CelebA-HQ we apply random left-right flipping whenever an image is loaded into a batch. For all the models we use $10\%$ of the training dataset as validation set, apart from CelebA-HQ for which we use the predefined train-val-test split.

## D.5  TRAINING PROCEDURE

On the MNIST datasets we train all models for $100K$ iterations, and we evaluate the test metrics on the iteration that achieved the best validation loss. For CelebA-HQ ($32 \times 32$ and $64 \times 64$), we train instead for $1M$ iterations and do early stopping if the validation loss does not improve for more than $20k$ iterations. ImageNet models are trained for $2M$ iterations with early stopping set to $100K$ iterations with no improvement. For generative modeling on CelebA-HQ, ImageNet, and all the denoising models we use gradient clipping if the magnitude of the gradients is bigger than 200. As optimizer we choose ADAM (Kingma & Ba, 2015) with a learning rate $1e - 3$ for the MNIST-like datasets, and $1e - 4$ for denoising experiments, CelebA-HQ and ImageNet. We use a batch size 128 for all the MNIST experiments, a batch size of 64 for CelebA-HQ resized to $32 \times 32$ and a batch size of 16 for CelebA-HQ resized to $64 \times 64$ and ImageNet.

We use *Weights & Biases* to track all our experiments and store models' checkpoints. (Biewald, 2020)

## D.6  IMPORTANCE SAMPLING

To get an estimate of the marginal likelihood for both VAE and AEF (linear) we use importance sampling with an average of 20 times 128 samples.

## D.7  COMPUTE RESOURCES

We used AzureML and Google Colab Pro to run all our experiments. All the manifold learning and denoising experiments are run on one GPU NVIDIA M60, while we use one GPU NVIDIA K80 for the experiments on CelebA-HQ and ImageNet.

## D.8 MODELS' PARAMETERS

We report the number of parameters used in each model for the different datasets: Table 3 for the models trained on MNIST-like datasets, Table 4 for CelebA-HQ resized to $32 \times 32$, and Table .

| Models | Nr. of latent dimensions | | | | |
|---|---|---|---|---|---|
| | 2 | 4 | 8 | 16 | 32 |
| VAE | 2.37M | 2.42M | 2.52M | 2.71M | 3.12M |
| AEF | 2.37M | 2.42M | 2.52M | 2.71M | 3.12M |
| AEF (linear) | 2.37M | 2.42M | 2.53M | 2.73M | 3.15M |

Table 3: Approximate number of model parameters for all models trained on MNIST-like datasets.

| Models | Nr. of latent dimensions | | |
|---|---|---|---|
| | 64 | 128 | 256 |
| VAE | 2.80M | 4.07M | 8.32M |
| AEF (linear) | 2.99M | 4.46M | 9.11M |

Table 4: Approximate number of model parameters for all models trained on CelebA-HQ and ImageNet rescaled to $32 \times 32$.

| Models | Nr. of latent dimensions | | |
|---|---|---|---|
| | 128 | 256 | 512 |
| VAE | 4.34M | 9.40M | 28.0M |
| AEF (linear) | 5.91M | 12.5M | 34.3M |

Table 5: Approximate number of model parameters for all models trained on CelebA-HQ rescaled to $64 \times 64$.

# E ADDITIONAL RESULTS

## E.1 GENERATIVE MODELING AND MANIFOLD LEARNING

In this section we provide additional results. Table 6 shows the results of all runs on CelebA-HQ and ImageNet. We show results comparing the linear and partitioned versions of AEF with VAEs on the MNIST datasets in Figure 5 and Figure 6. Figure 8 shows samples of VAEs and AEFs trained on MNIST, FashionMNIST and KMNIST with a latent dimensionality of 32, and Figure 9 shows the same for CelebA-HQ resized to $64 \times 64$ for various latent dimensionalities.

| Models | | CelebA ($64 \times 64$) | | | CelebA ($32 \times 32$) | | | ImageNet | |
|---|---|---|---|---|---|---|---|---|---|
| | | 64 | 128 | 256 | 64 | 128 | 256 | 128 | 256 |
| VAE | BPD | 6.98 | 6.87 | 6.79 | 6.98 | 6.70 | 6.63 | 6.89 | 6.76 |
| | | 6.99 | 6.86 | 6.79 | 6.90 | 6.64 | 6.65 | 6.92 | 6.74 |
| | FID | 106 | 96.3 | 90.9 | 91.9 | 68.4 | 63.5 | 179 | 164 |
| | | 102 | 92.6 | 92.1 | 87.6 | 67.0 | 66.2 | 182 | 160 |
| AEF | BPD | 6.35 | 6.07 | 5.78 | 6.20 | 5.86 | 5.58 | 6.30 | 6.24 |
| | | 6.35 | 6.07 | 5.79 | 6.21 | 5.93 | 5.57 | 6.31 | 6.21 |
| | FID | 76.5 | 60.8 | 55.8 | 51.9 | 38.4 | 27.9 | 131 | 123 |
| | | 76.2 | 62.6 | 54.9 | 53.3 | 40.2 | 30.9 | 132 | 124 |

Table 6: Bits per dimension (BPD) and Frechet Inception Distance (FID) for both runs of AEF and VAE models trained on CelebA and ImageNet. The second row denotes the latent dimensionality used. Lower is better for both metrics.

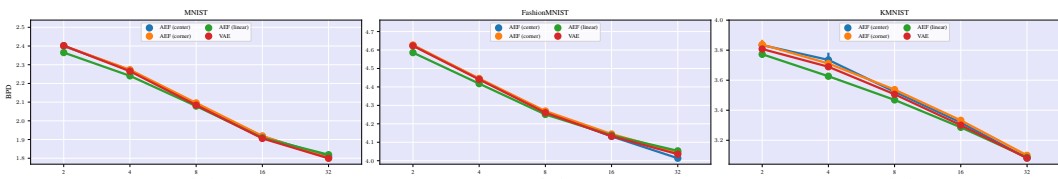

Figure 5: Bits per dimension achieved by AEFs and equivalent VAE MNIST, FashionMNIST and KMNIST. We show the mean and 95% confidence interval over 5 runs.

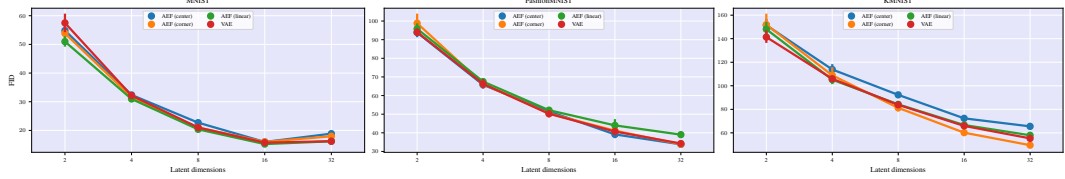

Figure 6: FID score achieved by achieved by AEFs and equivalent VAE on multiple datasets. We show the mean and 95% confidence interval over 5 runs.

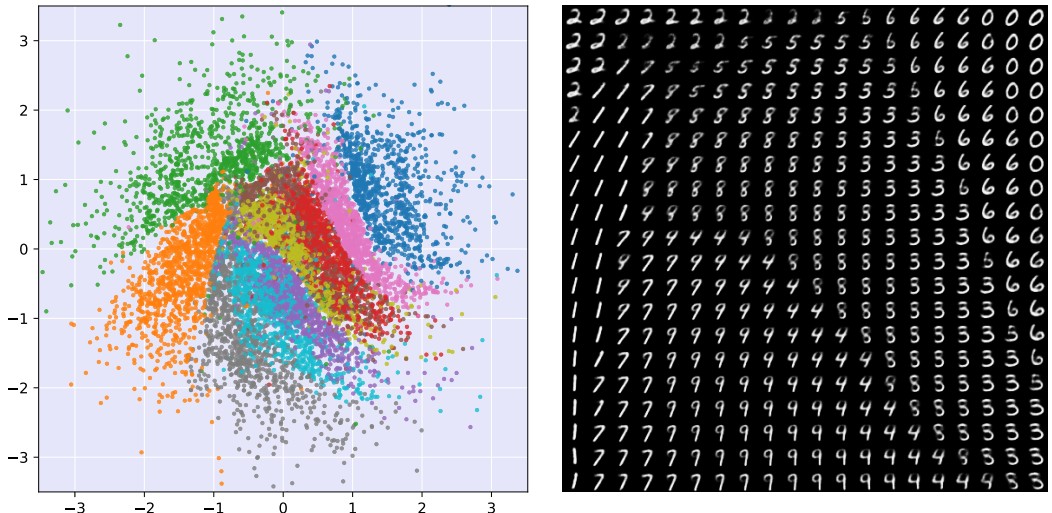

Figure 7: Visualization of a learned two-dimensional manifold by AEF trained on MNIST with a two-dimensional latent space.

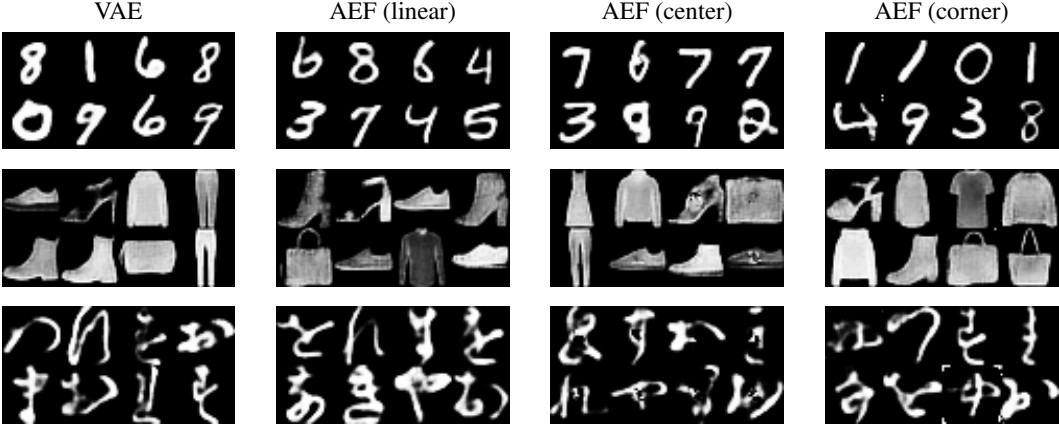

Figure 8: Samples generated by VAEs and AEFs trained on MNIST, FashionMNIST and KMNIST with 32 latent dimensions. Samples were generated with a temperature of 0.85.

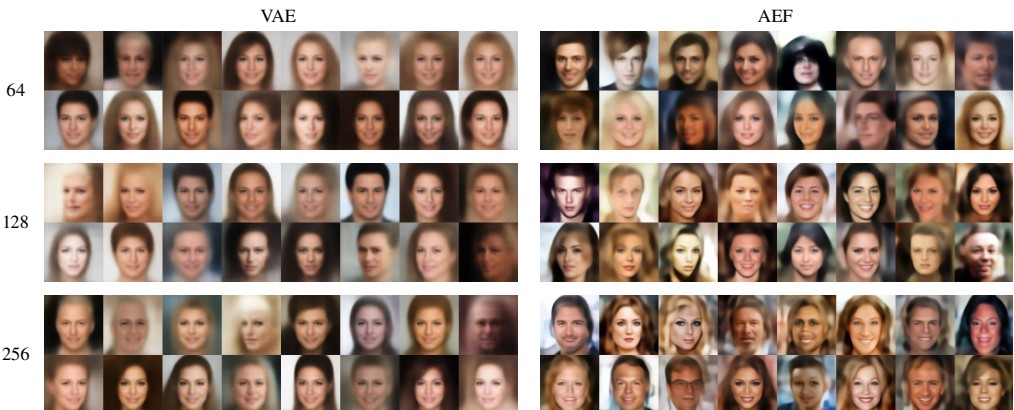

Figure 9: Samples generated by VAEs and AEFs trained on CelebA-HQ resized to $64 \times 64$ with 64, 128 and 256 latent dimensions. Samples were generated with a temperature of 0.85.

## E.2 Denoising

This section presents additional results and figures comparing the denoising performance of AEFs to baseline models, specifically to a VAE with equivalent architecture and a least squares denoising autoencoder (AE) with equivalent encoder and decoder. Figure 10 presents the mean squared error between inception feature activations (IFE) for increasing levels of noise MNIST, FashionMNIST and KMNIST datasets.

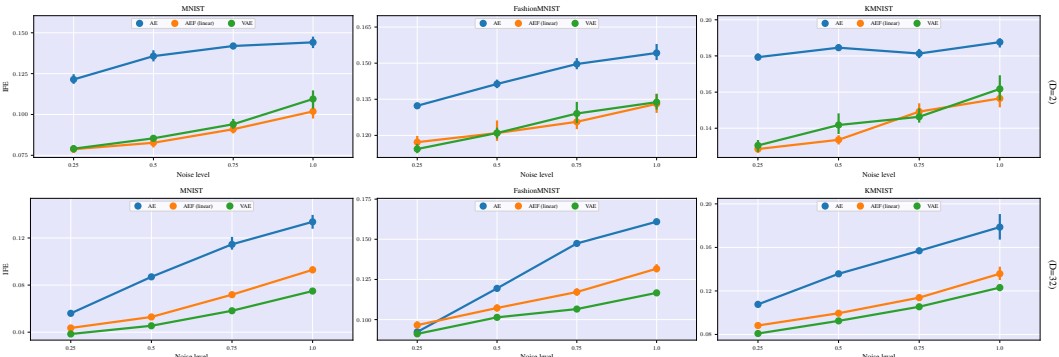

Figure 10: Mean squared error between inception feature activations of the original inputs, and the reconstructions based on a noisy input for increasing levels of noise. Averaged over five runs. Bars indicate 95% confidence intervals. The number of latent dimensions for was set to 2 for the upper row, and 32 for the lower row.

## F Ablations

In this section we investigate the importance of the prior and posterior flow on the performance of both VAEs and AEFs. We train a VAE and AEF with and without prior and posterior flow on CelebA-HQ ($32 \times 32$), and report the results in BPD and FID in Table 7. Qualitatively and by FID score we observe that for both VAE and AEF the quality of samples generated without a prior flow present in the model are of significantly worse quality than when there is a prior flow present. Additionally, having only a posterior flow decreases samples quality in both the VAE and the AEF. It is interesting to note that an AEF without core encoder and prior flow still performs better than a VAE with both posterior flow and prior flow in terms of BPD and FID score. Examples of samples for each ablation are presented in Figure 11 and Figure 12.

|  | BPD | FID |
|---|---|---|
| VAE (no flows) | 6.68 | 77.2 |
| VAE (only posterior) | 6.56 | 81.5 |
| VAE (only prior) | 6.71 | 74.1 |
| VAE (both flows) | 6.67 | 68.1 |
| AEF (no flows) | 5.90 | 64.4 |
| AEF (only posterior) | 5.88 | 108 |
| AEF (only prior) | 6.05 | 37.6 |
| AEF (both flows) | 5.90 | 35.6 |

Table 7: Ablation results of VAEs and AEFs with and without a posterior flow/core encoder and prior flow.

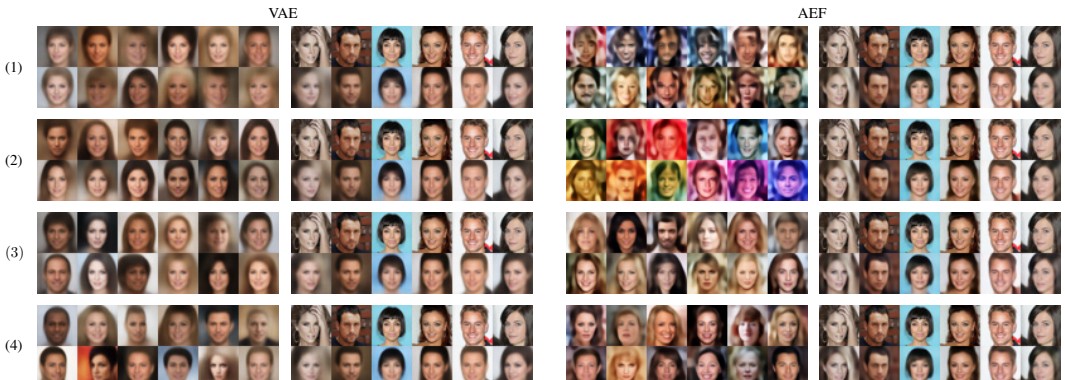

Figure 11: Samples and reconstructions of VAEs and AEFs trained on CelebA-HQ resized to $32 \times 32$ with: 1) no posterior or prior flow; 2) only a posterior flow; 3) only a prior flow; 4) both posterior and prior flows. All models had a latent dimensionality of 128. Samples were generated with a temperature of 0.85.

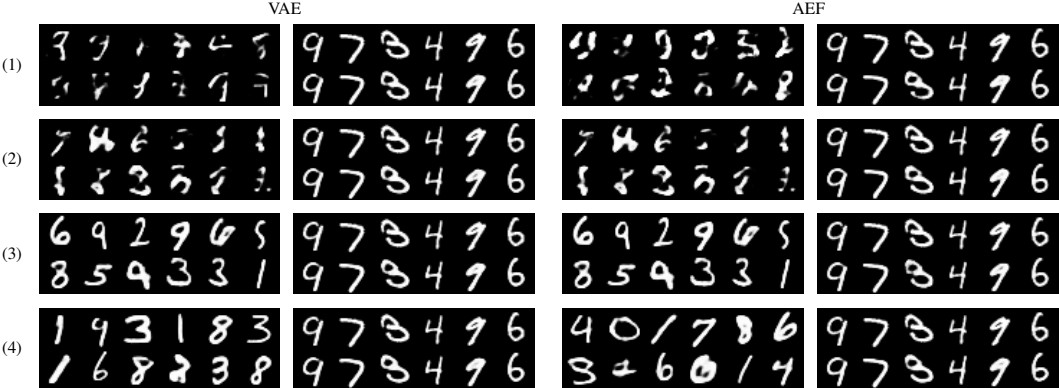

Figure 12: Samples and reconstructions of VAEs and AEFs trained on MNIST with: 1) no posterior or prior flow; 2) only a posterior flow; 3) only a prior flow; 4) both posterior and prior flows. All models had a latent dimensionality of 32. Samples were generated with a temperature of 0.85.

