# OpenReview forum: "Deterministic training of generative autoencoders using invertible layers"
_ICLR.cc/2023/Conference — ICLR 2023 notable top 25%_

### Official Review · Reviewer_jiVP · 2022-10-22

**Confidence:** 4
**Correctness:** 4
**Technical Novelty And Significance:** 3
**Empirical Novelty And Significance:** 3
**Recommendation:** 6

**Clarity, Quality, Novelty And Reproducibility:**

- Appendix A --- or at least a summary of this, should be included in the main paper. Is this a type of SurVAE Flow layer? They also use delta functions to describe exact log likelihood in one direction. The explanation in the appendix seems a little weak or vague.  For example, the constant in (12) is not explained---it is worth it to expand on this even though I think I understand. Can you expand on this and the intuition in the main paper?

- Figure 1 is definitely helpful for understanding how everything fits together. I wonder if it may be possible to introduce an outline of the approach earlier on to help guide the reader through the steps and through the two different approaches. It is hard to put it all together or see how all the pieces fit together.

- It would be helpful to write down the entire $\Phi$ function for both AEF models to easily verify invertibility and the overall strucure.

- Organization - Splitting the discussion of the partition and expanded AEFs would help clarify that these are two distinct approaches.  Also, clarifying the pros and cons of each would be helpful. I believe partition AEFs allow for exact log likelihood while expanded AEFs do not. But expanded AEFs are simpler to design and sample from since they do not require specifying the core and shell of the original input.

- Putting the VAE equation and the AEF equation directly next to each other would be helpful (and may be worth repeating the equation). Generally, more discussion on the similarities and differences between the objectives would be interesting.


**Strength And Weaknesses:**

*Strengths*
- The approach can use standard VAE and AE architectures directly within the framework. Thus, it provides an elegant bridge between VAEs and flows.

- The expanded AEF version does not require specifying the shell and core variables which makes it more broadly applicable as a dro-in replacement for standard VAEs.

*Weaknesses*
- How does your work compare to SurVAE Flows [Nielsen et al., 2020]? This work also bridges VAEs and flow models. Can your model be cast as a type of SurVAE Flow?  This seems to be a significant oversight in related works and may suggest another reasonable baseline.

Nielsen, D., Jaini, P., Hoogeboom, E., Winther, O., & Welling, M. (2020). Survae flows: Surjections to bridge the gap between vaes and flows. Advances in Neural Information Processing Systems, 33, 12685-12696.


- (Minor) It is not clear that the sampling procedure for estimating the likelihood for the expanded AEF is good, and it may be impossible to check. While the VAE values are a true lower bound on log likelihood. The FID and qualitative samples definitely help support the overall claims so this is not necessarily a huge concern but it should be noted that the sampling procedure is merely an approximation to the likelihood and could be above or below the true likelihood.


**Summary Of The Paper:**

The paper proposes to overcome the training limitations of VAEs as both manifold learners and generative models.
Essentially, the proposed AEF architecture is one giant coupling layer.
It transforms some dimensions (called the "core") using a standard normalizing flow (similar to the use of inverse autoregressive flows for latent variables) and conditions on the other dimensions (called the "shell").
The key trick to make the whole architecture invertible is to save the residuals of the reconstructed x as additional variables (i.e., the other latent dimensions for a flow model).
Thus, the core latent space + the residuals of the shell space is the entire space and can be inverted.
The paper extends this to allowing an augmented space where the augmentation variables are functions of the input, and while this disables observed log likelihood computation, it avoids having to choose the shell vs core.
Finally, the paper shows that this new approach is usually better than the equivalent VAE in terms of FID, bits per dimension, and qualitative inspection of samples and reconstructions.


**Summary Of The Review:**

This paper proposes an simple yet useful combination of autoencoder ideas and normalizing flows to enable flexible VAE-like normalizing flows that could be used as drop-in replacements for standard VAEs. Overall, this is an excellent direction and a nice contribution. The main weakness is the lack of comparison to SurVAE flows, which seem to have similar ideas and bridge VAEs and flows more fully. Though I believe the contributions are still interesting especially by making it a drop-in replacement for VAEs, I believe the proposed technical ideas are less novel because of the prior work in SurVAE flows. A good comparison to this prior work could improve my score.

---

> ### Author Response · Authors · 2022-11-11
> **SurVAE reference and importance sampling**
>
> We thank the reviewer for the insightful and detailed feedback. We are pleased that you found our work to be useful, elegant and novel.
>
> In the following, we will address some of the issues and questions.
>
> ## Missing reference
> We agree that both our method and SurVAE combine normalizing flows and VAE techniques. We did include the paper in the revised related work section. However, as we explained in the revision, the contributions of the two works are almost orthogonal as they do solve different problems using different methods. In this sense, the similarities are rather superficial.  SurVAE uses training strategies from VAEs to let flows model surjective transformations, which would not be possible with traditional normalizing flows. These include lossy transformations such as quantization, discretization and downsampling. The resulting model does not have a tractable likelihood and needs to be trained using the ELBO like a VAE. The main goal of our work is instead to use invertible layers to train generative autoencoders by exact maximum likelihood, without using the ELBO and stochastic gradient estimation during training. This can be similarly seen as a normalizing flow that is capable of dimensionality reduction, which is not the goal of SurVAE but is instead more closely related to the injective/rectangular flow literature. In fact, SurVAE layers can be incorporated in AEF models to account for surjective transformation, for example for modeling discrete data. On the other hand, AEF techniques can be added to SurVAE models to perform dimensionality reduction (or expansion).
>
> ## Evaluation metric
> Our importance sampling likelihood estimation method used to compute the BPD values is conceptually identical to the one used for the VAEs and commonly used in the literature. In fact, while the analytic ELBO is a lower bound, the stochastic estimate that it is used in practice is not an exact lower bound. Moreover, since the ELBO is biased, BPD values are estimated using importance sampling to reduce the bias. Therefore, while we agree that it is important to keep the stochastic properties of the estimation methods in mind, this is also true for the results of the VAEs. In both cases, the likelihood estimates are unbiased, although the (true) variance can be high due to outliers. In practice, the variance of both estimators is usually small compared with the magnitude of the mean values.

---

> > ### Comment · Reviewer_jiVP · 2022-11-16
> > **Some disconnects or misunderstandings remain**
> >
> > Hi authors,
> >
> > Thank you for your response. I believe you have at least partially misunderstood SurVAE flows, which is concerning.  One surjective direction of SurVAE flows can be trained with exact likelihood (i.e., there is no bound gap) despite your claim that "The resulting model does not have a tractable likelihood and needs to be trained using the ELBO like a VAE".  (The other surjective direction does have a bound gap for training.) Additionally, you claim that "dimensionality reduction, which is not the goal of SurVAE".  Again, while it may not be the only goal of SurVAE, SurVAE does indeed have a dimensionality reduction method as its first example, see "Example 1: Tensor slicing". While I agree that it is not the same objective as your work, both of these confusions suggest that you have not deeply understood SurVAE flows and is concerning.
> >
> > For the second part, I'm not convinced that this is the same as ELBO.  For ELBO there are two distinct parts: 1) It is a lower bound theoretically, and 2) it is approximated with samples.  These are two different things.  Your argument that both are stochastic is true.  However, I do not believe your importance sampling method can be proved to be a lower bound as well. Thus, it is still fundamentally different and could favor your method over the VAE methods. Please be explicit about why it is a lower bound if you believe it is indeed a lower bound as well.

---

> > > ### Author Response · Authors · 2022-11-16
> > > **Further clarifications**
> > >
> > > Dear reviewer,
> > >
> > > Thank you for the quick reply. In the following, we will attempt to clarify the remaining issues. We updated the submission to make the comparison with SurVAE clearer. The new version includes an extensive treatment of SurVAE and of its connection with AEFs (See end of section 4). We hope you will find this more extensive discussion informative.
> > > ## Clarifications about SurVAE
> > > We are glad you agree that the SurVAE flow paper has a different objective than our work. We would like to state the difference clearly, our work uses a regular flow architecture in a peculiar way to perform generative autoencoding, while SurVAE generalizes flow architectures using surjective layers, which go beyond the standard flow theory. However, as we explained in the new revision, the surjective generalisation does not offer a straightforward way to perform generative dimensionality reduction and autoencoding, which in fact is not discussed in the original paper. So said, we do agree that a clearer explanation of SurVAE flows will make the manuscript stronger and to this aim, we included a whole new paragraph at the end of the related work section covering it in detail.
> > > ## Evaluation metrics
> > > We are not making any claim about the equivalence of our importance sampling estimate and the ELBO. Both for VAEs and AEFs, for the purpose of evaluation, the likelihood is estimated using importance sampling. Both estimators are provably unbiased. The use of importance sampling for estimating BPD values is standard in the VAE and flow literature. Neither of the estimators used for computing the numbers in the table are lower bounds. In fact, it would be inappropriate to report the lower bound as it systematically underestimates the likelihood. Since they are both forms of unbiased importance sampling, we do not believe that the estimators are fundamentally different. If anything, the comparison is more likely to favour the VAE since its sampler is an IAF trained to maximize the ELBO while ours follows a fixed spherical Gaussian distribution.

---

> > > > ### Comment · Reviewer_jiVP · 2022-11-17
> > > > **Thanks for the additional clarification**
> > > >
> > > > Thank you for the response. I appreciate the fuller discussion of SurVAE flows. For the evaluation, thank you for the clarification that you used importance sampling for both VAEs and AEFs. I had mistakenly assumed you used the ELBO for VAEs, which is also fairly common in the literature to say that the likelihood >= ELBO value. Also, Appendix A only talks about importance sampling for AEFs (linear) which added to the confusion. Later I noticed that this is buried in Appendix B.6. Please add this to the caption of the table and mention briefly in experiments section to make this clear.

---

> > > > > ### Author Response · Authors · 2022-11-17
> > > > > **Clarification about importance sampling added in the caption of the table**
> > > > >
> > > > > Dear reviewer,
> > > > >
> > > > > We are happy that you appreciated our revision and that everything is now clear concerning the evaluation metrics. As suggested, we added a reference to appendix B.6 to the caption of the table in the experiment sections, together with a short explanation stating that importance sampling is used for estimating BPD in both VAE and AEF.

---

### Official Review · Reviewer_4VSE · 2022-10-23

**Confidence:** 4
**Correctness:** 4
**Technical Novelty And Significance:** 3
**Empirical Novelty And Significance:** 2
**Recommendation:** 8

**Clarity, Quality, Novelty And Reproducibility:**

**Clarity**
I believe the paper's overall clarity could be improved by merging subsections 3.1-3.4 into fewer sections.  Since the actual model proposed uses the expanded ambient space, the partitioning idea could be skipped altogether.  It may be helpful to explicitly state the two different interpretations of the model next to each other: (1) as a flow model on the expanded ambient space with the expanded dimensions corresponding to the "prior", and the original ("shell") dimensions corresponding to the reconstruction error, and (2) as a VAE model with deterministic encoder with Gaussian decoder (though I understand that a different choice of error distribution can lead to non-Gaussian decoder).

**Quality**
Besides perhaps the verbose writing style, this is a high quality paper.  The idea presented is sensible and is followed by a thorough discussion.  I particularly appreciated the fact that the authors listed current limitations of the proposed approaches in Sec 6.

**Originality**
I believe the idea proposed in the paper is novel.

**Strength And Weaknesses:**

**Strength**
* Interesting idea with clearly explained motivations.
* Clear empirical results, both qualitative and quantitative.
* Thorough discussion on how the proposed method relates to existing approaches.

**Weakness**
* Title is misleading -- the proposed method does not perform max-likelihood training of a VAE with an arbitrary architecture.  It instead constructs a normalizing flow model in the expanded ambient space and trains that flow model with exact likelihood instead.  See my first question below.
* Writing is a bit verbose and could be condensed (see "Clarity" below).

**Questions**
* It's unclear to me how the ambient space expansion affects the likelihood.  Authors point out in Sec 3.4 that $p(x)$ does not have a particular advantage over $p(x, w)$ under the manifold hypothesis, but I believe this point is debatable. More importantly, it would be informative to know how $p(x, w)$ relates to the tightness of the ELBO.
* I'm curious what the learned error distribution $r(\delta; \theta)$ looks like.  Speficially: how does actually sampling $\delta \sim r()$ change the samples, and what is the learned value of $\sigma$?

**Minor comments**
* Eq. 4 uses $g_{\sigma}$ but the text uses $g_s$


**Summary Of The Paper:**

The paper proposes a novel scheme for training a VAE by embedding it inside a normalizing flow model on the expanded ambient space.  Specifically, the method uses an invertible network to transform the expanded ambient dimensions followed by an affine coupling layer to compute the "latent" variable of the embedded VAE.  To satisfy the invertibility requirement, the output of the VAE decoder is augmented with error (residual) terms.  Authors then empirically show that this flow-VAE hybrid scheme leads to qualitatively sharper images and improved log likelihood on MNIST, CelebA-HQ and ImageNet 32x32.

**Summary Of The Review:**

The paper proposes a novel training scheme for VAE by embedding it inside a flow model in an expanded ambient space. The idea is well-motivated, and the empirically results show clear improvement.  I believe the proposed method is an interesting hybrid of VAEs an normalizing flows, and is valuable to the community.

---

> ### Author Response · Authors · 2022-11-11
> **Sampling from deviations and likelihood in the extended space**
>
> We wish to thank the reviewer for the detailed and positive review. We agree that the clarity of the paper can be improved with some adjustments and we incorporated the suggestions in the new revised version. However, we decided to keep the partition model in since 1) it has the advantage of having its likelihood defined on the original space and 2) it makes it easier to understand the model on the expanded space. Note that we did run experiments for the partition model on MNIST, fMNIST and kMNIST.
>
> In the following, we will address some of the remaining questions:
> ## Sampling from the deviations
> Sampling from the deviations has the only effect of adding white noise to the samples. There is usually no benefit in doing so as it will corrupt the generated image. Our noiseless sampling approach is analogous to what is commonly done for VAEs.
> ## Likelihood in the extended space
> Under the manifold hypothesis, the data is (at least locally) generated by a latent distribution p(z) and then mapped to the ambient space through an injective transformation that depends on the measuring device. In our dimensionality expansion approach, we simply apply another injective transformation. These two transformations can be composed, resulting in a single injective mapping applied to the latent variables. This preserves the dimensionality of the signal and the assumptions behind the manifold hypothesis. In this sense, at least in this regime, the original likelihood p(x) is as valid as a training loss as p(x, w). In fact, under the manifold hypothesis, the value of p(x) is rather arbitrary, as for example explained in (Loaiza-Ganem, 2022).
> Concerning the relationship with the tightness of the ELBO, this is difficult to answer for us as our method does not use an ELBO and we are not aware of a clear mapping between our likelihood and the tightness of an ELBO in an architecturally equivalent VAE.
>
> ### Citations
>
> Loaiza-Ganem, Gabriel, et al. "Diagnosing and Fixing Manifold Overfitting in Deep Generative Models." arXiv preprint arXiv:2204.07172 (2022).

---

> > ### Comment · Reviewer_4VSE · 2022-11-16
> > **Response**
> >
> > Thanks for the response.
> >
> > Regarding likelihood, I'm a bit confused by the overall stance of the paper.  As the *title* of the paper emphasizes the fact that training is done via exact maximum likelihood, it gives me the impression that likelihood is a meaningful objective.  But at the same time the authors state that "In fact, under the manifold hypothesis, the value of $p(x)$ is rather arbitrary".
> >
> > I'm not looking to have a debate on whether marginal likelihood $p(x)$ is meaningful or not in the context of VAE training.  However, I do believe that the paper shouldn't advocate the proposed method for its ability to perform exact likelihood training, and at the same time advocate that it's okay to maximize $p(x, w)$ because $p(x)$ is not very meaningful anyway.
> >
> > On a related note, I still find the title very misleading and needs to be fixed -- exact likelihood training should maximize $p(x)$, not $p(x, w)$.

---

> > > ### Author Response · Authors · 2022-11-16
> > > **Change of title and some clarifications on the nature of the ambient space likelihood**
> > >
> > > Dear reviewer,
> > >
> > > Thank you for the reply. We agree that there is a tension in the overall stance of some parts of the paper that can be evened out in order to make things more clear. As explained below, we will do that by making some textual adjustments and by changing the title.
> > >
> > > We propose:
> > > Deterministic training of generative autoencoders using invertible layers.
> > >
> > > Concerning the exact likelihood claim, we would like to stress that the partition model we provided does provide the exact ambient-space likelihood p(x). The experiments in the Supplementary show that these non-extended models perform similarly to the extended model. In the revised version, we will therefore de-emphasize the claim and make more clear that it refers to the partition model.

---

> > > > ### Comment · Reviewer_4VSE · 2022-11-17
> > > > **Response**
> > > >
> > > > Thanks for the prompt response.  The proposed title (or some variation of it as the authors see fit) seems good to me.  And thank you for clarification regarding the partition model.

---

### Official Review · Reviewer_Mtne · 2022-10-24

**Confidence:** 4
**Correctness:** 3
**Technical Novelty And Significance:** 4
**Empirical Novelty And Significance:** 4
**Recommendation:** 8

**Clarity, Quality, Novelty And Reproducibility:**

The paper is well structured and clearly written. The proposed model is to my knowledge novel.

**Strength And Weaknesses:**

Paper strengths:
The idea to enlarge the ambient space by additional variables that can be interpreted as manifold coordinates and then applying a normalising flow to these base coordinates combined with a second invertible mapping that accounts for the remaining part of input, seems to be well suited for the considered assumptions. The remaining part of the latent code can be then interpreted as independent residual noise. The price is however, that parts of the decoder are needed for the full formulation of the encoder.

The experiments show that the resulting model provides better results than VAEs with comparable encoder/decoder architectures.

Paper weaknesses:
I would expect that the model can be better justified (ab initio) by starting from appropriate assumptions about the distribution $p(x,w)$, where $w$ denotes the additional dimensions of the enlarged ambient space.

In order to learn the parameters of the feature expansion mapping by maximising likelihood, the authors propose to interpret it as a deterministic limit of a normal distribution. This and the requirement to assume distributions with learned scale for the residual part of the latent variables are in my view conceptually not fully convincing.

**Summary Of The Paper:**

The paper proposes a novel generative model class for situations when the data distribution is concentrated around a low dimensional manifold in the input space. The authors propose to enlarge the input space by the outputs of a non-linear parametrised mapping that can be thought to represent the manifold coordinates of a data point. They then define a normalising flow model on this enlarged space, which leads to a deterministic invertible encoder. The corresponding decoder/encoder pair can be learned by maximising the data likelihood. Experiments show that the proposed model outperforms VAEs with comparable architectures on datasets like CelebA and ImageNet.

**Summary Of The Review:**

The paper proses a novel type of generative models under the assumption that the data distribution is concentrated around a low dimensional manifold in the input space. The derived likelihood maximisation seems to be correct and outperforming comparable VAEs learned by ELBO. It is perhaps possible to achieve a better ab-initio justification of the approach by starting from appropriate structural assumptions for the distribution $p(x,w)$ on the enlarged ambient space.

---

> ### Author Response · Authors · 2022-11-11
> **Residual scale and expansion parameters**
>
> We thank the reviewer for the positive review.
> In the following, we will address some of the issues and questions.
>
> ## Training the expansion parameters
> We double-checked the derivation of the loss as a limit of KL divergences and we did not find any error. However, we agree that our derivation in the original submission was short and difficult to understand. Therefore, we moved Appendix A to the main text (section 3.3) and expanded it. In this new version, we better explained the motivation behind the KL loss between the joint distributions and the details of the deterministic limit.
>
> ## Learnable residual scale
> We do not require the residual to have a learnable scale parameter. However, using learnable scale parameters leads to substantially higher performance as the model can adjust to the scale of the residuals, which tends to decrease during training. Note that, at least in our experiments, using a learnable residual scale is also very beneficial for the VAEs.

---

> > ### Comment · Reviewer_Mtne · 2022-11-18
> > **Response**
> >
> > Thank you for the comments. I believe that including Appendix A into the the main text and expanding it, has improved the readability of the technical part of the paper. I am keeping my recommendation to accept the paper.

---

### Official Review · Reviewer_LqjZ · 2022-10-25

**Confidence:** 4
**Correctness:** 4
**Technical Novelty And Significance:** 3
**Empirical Novelty And Significance:** 2
**Recommendation:** 8

**Clarity, Quality, Novelty And Reproducibility:**

Clarity: Generally clear

Quality: See strengths/weaknesses

Novelty: Ideas are novel AFAIK

Reproducibility: Seems very reproducible - code is provided along with notebooks

**Strength And Weaknesses:**

Strengths:
 - A conceptually interesting method and integration of VAEs with normalising flows
 - Clear and well-structured

Weaknesses:
 - The main weakness of this paper is that the architectures experimented with are far from the state-of-the-art, coming from (Kingma and Welling, 2014). Despite having referenced "compatibility with the VAE literature", and specifically VAEs with higher-dimensional latent spaces than data spaces, as a motivation for the ambient space expansion, none of the VAEs in the experiments have this property. I would be interested in e.g. whether this method could be applied in some way to more modern (e.g. hierarchical) VAE architectures.
- Based on Figure 10 and Table 2, it seems like the advantage of AEFs for denoising only holds for low latent dimensionality, and diminishes as the latent dimensionality increases. This suggests that this method is unlikely to be useful for practical denoising tasks. I would also be worried about whether the same scaling with latent dimensionality holds for other tasks (e.g. use as a generative model).
- Another possible advantage of older VAE architectures, mentioned by the authors, is the "interpretability" of their latent space and "their ability to project complex data into a semantically meaningful set of latent variables", but the authors do not demonstrate that the interpretability of their method's latent features (or usefulness for downstream tasks) is any better than that of baselines. I mention this merely because, based on my previously listed weaknesses, the method does not appear to have significant advantages as a generative model or as a denoiser. I would see this as a much stronger paper if the authors could make a convincing case for their method's applicability to at least one use-case.

Minor:
 - Acronym "AEF" used in paragraph afer Eq. 5 without having been previously introduced.

Other minor comment:
 - It is not clear to me what properties the learned auxiliary variables ($w=h(x; \gamma)$) would have. Some intuition about what makes a good auxiliary variable would be interesting.

**Summary Of The Paper:**

The authors propose a method to train an auto-encoder in which the mapping to a latent space is invertible, making the (exact) likelihood tractable and so allowing maximum likelihood training. The method involves augmenting the data with additional variables (obtained from a learnable function of the data) and then running a single affine normalizing flow layer on the augmented space. This normalizing flow layer encodes these additional variables using an affine function that depends on the original data and sets the remaining latent variables to be the residual of the data after subtracting the prediction given the latents. The prior in the latent space is Gaussian, and a learnable prior for the residuals encourages the network to drive the residuals to zero and therefore use the other latents.

In my mental model of this, the prior for the residuals has the feel of a pixelwise Gaussian likelihood in a traditional VAE. A significant difference from standard VAEs, though, is that no noise is added by the encoder.

**Summary Of The Review:**

The proposed method is interesting and demonstrates advantages over the VAEs introduced in (Kingma and Welling, 2014). The paper could be improved further with better demonstration of use-cases for this method, or demonstration of compatibility with more modern (e.g. hierarchical) VAE architectures.

---

> ### Author Response · Authors · 2022-11-11
> **AEF use cases**
>
> We want to thank the reviewer for the detailed and helpful review. In the following, we shall argue that our work supports several important use cases for AEF models.
>
> ## Use cases
> We believe we have made a convincing case for the advantages of AEF in the low-dimensional regime, observing especially significant improvements in the more complex datasets. Dimensionality reduction is very relevant for important applications such as compression [1,2], and model-based reinforcement learning [3,4,5], where a compact state representation leads to more efficient training. In any situation where a (variational) autoencoder is used to compress complex data into a condensed, lower-dimensional representation, our method may yield large improvements without adding additional architectural complexity. Note that the sub-optimal behaviour of VAE in low dimension (e.g. blurry samples) is an important open problem in the literature, with multiple high-profile papers aimed at ameliorating these issues (e.g. [6] and [7]).
>
> Indeed, more research needs to be done in order to compare VAEs and AEFs with more complex hierarchical architectures and higher dimensional regimes. However, scaling autoencoders to these regimes requires a substantial amount of fine-tuning and, in our opinion, goes beyond the scope of this paper.
>
> ### Citations
>
> [1] Bora, A., Jalal, A., Price, E. &amp; Dimakis, A.G.. (2017). Compressed Sensing using Generative Models. Proceedings of the 34th International Conference on Machine Learning, in Proceedings of Machine Learning Research
>
> [2] Gregor, K., Besse, F., Jimenez Rezende, D., Danihelka, I., & Wierstra, D. (2016). Towards Conceptual Compression. In Advances in Neural Information Processing Systems. Curran Associates, Inc..
>
> [3] Ha, David, & Schmidhuber, Jürgen. (2018). World Models. https://doi.org/10.5281/zenodo.1207631
>
> [4] Danĳar Hafner, Timothy Lillicrap, Jimmy Ba, & Mohammad Norouzi (2020). Dream to Control: Learning Behaviors by Latent Imagination. In International Conference on Learning Representations.
>
> [5] Luisa Zintgraf, Kyriacos Shiarlis, Maximilian Igl, Sebastian Schulze, Yarin Gal, Katja Hofmann, & Shimon Whiteson (2020). VariBAD: A Very Good Method for Bayes-Adaptive Deep RL via Meta-Learning. In International Conference on Learning Representations.
>
> [6] Rezende, Danilo Jimenez, and Fabio Viola. "Taming vaes." arXiv preprint arXiv:1810.00597 (2018).
>
> [7] Dai, Bin, and David Wipf. "Diagnosing and enhancing VAE models." arXiv preprint arXiv:1903.05789 (2019).

---

> > ### Comment · Reviewer_LqjZ · 2022-11-17
> > **Thanks for the response**
> >
> > Thanks for the response. Given this and the other changes described I have raised my score to an 8

---

### Author Response · Authors · 2022-11-11
**Common answer**

We thank all the reviewers for their insightful reviews. We are very pleased to see that all reviewers agree that the paper should be accepted, based on the relevance and innovation of our method, as well as on the overall quality and clarity of the manuscript.

We uploaded a new version of the paper that addresses several of the reviewers' observations, comments and questions. The main change is that we moved Appendix A to the main text (section 3.3) and we expanded its content. We also included a comparison with the SurVAE approach in our related work section.

We further address more specific feedback in separate replies to the corresponding reviewers.

[UPDATE 17/11]
Based on the comments made by 4VSE, we decided to change the title of the manuscript to make it more descriptive of the main content of the paper. Furthermore, we made several textual changes and restricted our claim of exact likelihood only to the partition AEF model. We thank the reviewer for the important inpu, which we think improved the overall clarity of our work.

The updated title is: Deterministic training of generative autoencoders using invertible layers

---

### Decision · Program_Chairs · 2023-01-20

**Decision:**

Accept: notable-top-25%

**Justification For Why Not Higher Score:**

It's a solid paper, but the experimental results are not spectacular.

**Justification For Why Not Lower Score:**

It's a solid paper. See meta-review.

**Metareview: Summary, Strengths And Weaknesses:**

Ratings: 8/8/8/6
Confidence: 4/4/4/4
Recommendation: Accept

This paper proposes an autoencoder-like normalizing flow that makes sense when the data distribution is on a low-dimensional manifold. The model is optimized by maximizing the log-likelihood, and learns a low-dimensional latent space from which the full data may be (approximately) reconstructed. There are two versions of the model: with input partitioning; or with an expanded ambient space. In the latter case, sampling is especially simple, and requires evaluating only an decoder network.

The model is a conceptually interesting integration of VAEs with normalising flows, and the paper is clear and well-structured.

Some reviewers had concerns regarding the lack of strong experimental results. However, there was a fruitful discussion with multiple back-and-forths, after which multiple reviewers upgraded their scores.


**Note From Pc:**

if the above contains the word "oral" or "spotlight" please see: "oral" presentation means -> notable-top-5% and "spotlight" means -> notable-top-25%. As stated in our emails, we are disassociating presentation type from AC recommendations